# Results of DUET: A Web-Based Weight Loss Randomized Controlled Feasibility Trial among Cancer Survivors and Their Chosen Partners

**DOI:** 10.3390/cancers15051577

**Published:** 2023-03-03

**Authors:** Wendy Demark-Wahnefried, Robert A. Oster, Tracy E. Crane, Laura Q. Rogers, W. Walker Cole, Harleen Kaur, David Farrell, Kelsey B. Parrish, Hoda J. Badr, Kathleen Y. Wolin, Dori W. Pekmezi

**Affiliations:** 1Department of Nutrition Sciences, University of Alabama at Birmingham (UAB), Birmingham, AL 35294, USA; 2O’Neal Comprehensive Cancer Center at UAB, Birmingham, AL 35233, USA; 3Department of Preventive Medicine, UAB, Birmingham, AL 35233, USA; 4Department of Medical Oncology, University of Miami, Miami, FL 33124, USA; 5Department of Health Behavior, UAB, Birmingham, AL 35294, USA; 6People Designs, Inc., Durham, NC 27705, USA; 7Department of Medicine, Baylor College of Medicine, Houston, TX 77030, USA; 8Coeus Health, LLC, Chicago, IL 60657, USA

**Keywords:** neoplasms, survivors, diet, exercise, weight reduction programs, internet, randomized controlled trial

## Abstract

**Simple Summary:**

Effective and scalable diet, exercise, and weight management interventions are needed for primary cancer prevention in the general public, as well as for cancer control and tertiary prevention among the growing population of cancer survivors. A 6-month online intervention, entitled “Daughters, dUdes, mothErs, and others Together” (DUET), was designed to promote weight loss, a healthful diet, and increased physical activity among cancer survivors and their chosen partners. Fifty-six cancer survivor-partner dyads (*n* = 112 participants in total) were recruited into a randomized controlled trial that compared DUET to a waitlist control. The trial surpassed all feasibility endpoints with regard to uptake, retention, and safety; the DUET intervention also resulted in significant weight loss and reductions in caloric intake, as well as having a promising impact on physical activity and performance, blood glucose, and indicators of inflammation.

**Abstract:**

(1) Background: A healthful diet, regular physical activity, and weight management are cornerstones for cancer prevention and control. Yet, adherence is low in cancer survivors and others, calling for innovative solutions. Daughters, dUdes, mothers, and othErs fighting cancer Together (DUET) is a 6-month, online, diet-and-exercise, weight-loss intervention to improve health behaviors and outcomes among cancer survivor-partner dyads. (2) Methods: DUET was tested in 56 dyads (survivors of obesity-related cancers and chosen partners) (*n* = 112), both with overweight/obesity, sedentary behavior, and suboptimal diets. After baseline assessment, dyads were randomized to DUET intervention or waitlist control arms; data were collected at 3- and 6-months and analyzed using chi-square, *t*-tests, and mixed linear models (α < 0.05). (3) Results: Retention was 89% and 100% in waitlisted and intervention arms, respectively. Dyad weight loss (primary outcome) averaged −1.1 (waitlist) vs. −2.8 kg (intervention) (*p* = 0.044/time-by-arm interaction *p* = 0.033). Caloric intake decreased significantly in DUET survivors versus controls (*p* = 0.027). Evidence of benefit was observed for physical activity and function, blood glucose, and c-reactive protein. Dyadic terms were significant across outcomes, suggesting that the partner-based approach contributed to intervention-associated improvements. (4) Conclusions: DUET represents a pioneering effort in scalable, multi-behavior weight management interventions to promote cancer prevention and control, calling for studies that are larger in size, scope, and duration.

## 1. Introduction

Over the past four decades, the prevalence of obesity has tripled [1], as has the number of cancer survivors [2]. While these trends are not directly related, obesity is an acknowledged risk factor for 13 different cancers [3]. Obesity and weight gain after a cancer diagnosis also are linked to a poorer prognosis [4]. Thus, weight management has been suggested as a means of primary cancer prevention [5], as well as tertiary prevention of second cancers and other prevalent forms of comorbidity among cancer survivors [6].

Given that social support plays an integral role in achieving the lifestyle changes that underlie weight management [7], buddy systems have been implemented in weight loss programs to enhance efficacy [8]. With respect to cancer, buddy pairings that unite cancer survivors with individuals in their social networks and which emphasize weight management to prevent or control cancer may be one way to expand reach and fortify intervention uptake, especially those that are minimal touch.

To date, there have been two dyadic interventions that have promoted diet, physical activity, and weight loss as a means of cancer prevention and control among both cancer survivors and their family members. The Daughters and Mothers (DAMES) study included 68 breast cancer survivors (mothers) who had overweight or obesity and were insufficiently active, paired with their adult biological daughters who had similar lifestyle behaviors and increased adiposity. Randomized dyads received 12 months of bimonthly print materials that were either team-tailored, individually tailored, or standardized (not tailored at all) [9]. Although baseline-to-follow-up improvements in weight, waist circumference, and accelerometry-measured physical activity were observed across all arms, as hypothesized, the tailored interventions resulted in significantly greater improvements in waist circumference and physical activity than the standardized intervention. Surprisingly, only the individually or personally tailored intervention and not the team-tailored intervention (which provided tailored feedback for both the survivor and daughter simultaneously) resulted in significant reductions in body mass index ([BMI] among cancer survivors but not daughters. While DAMES was deemed feasible in terms of safety, retention (90%), and achievement of the accrual target, recruitment of mother-daughter dyads was difficult (3% enrollment rate [#consented/#contacted]).

Subsequently, “Healthy Moves” focused on survivor-spouse dyads (*n* = 22), which were randomized to intervention arms that targeted the survivor alone (using a modified version of the DAMES individually tailored intervention) or both the survivor and spouse (using a modified version of the DAMES team-tailored intervention, plus nine videoconferencing sessions with a marriage counselor to enhance spousal communication) [10]. Compared to DAMES, Healthy Moves had a higher enrollment rate (12.7%); however, despite intensive efforts to enhance communication among couples assigned to the spouse-survivor arm, improvements in weight and fruit and vegetable consumption (and related physical function outcomes) were similar across arms. Spouses receiving the intervention versus spouses of dyads randomized to the survivor-alone condition (i.e., who did not receive the intervention) experienced significant improvements in lifestyle behaviors and physical function outcomes.

Building on this research, the current study, Daughters, dUdes, mothers, and othErs fighting cancer Together (DUET) trial, sought to expand eligibility by encouraging survivors to select any partner they felt could participate with them in the weight loss intervention [11]. Furthermore, instead of relying on a tailored, mailed print intervention that was informed by mailed surveys, DUET incorporated newer technology, i.e., Fitbits^®^ and Aria^®^ scales, for monitoring and used a more contemporary and scalable web-based platform to deliver the intervention. Herein, we report the main outcomes of the DUET trial that was aimed at promoting weight loss among cancer survivors and their chosen supportive partners. Hypotheses were that dyads assigned to the DUET intervention would lose significantly more weight (primary outcome) at 6-month follow-up than dyads assigned to the waitlisted control; moreover, the intervention also would result in more favorable changes in secondary outcomes, such as other measures of adiposity (e.g., waist circumference), diet quality, physical activity, quality-of-life, and physical performance, as well as related biomarkers (e.g., insulin, glucose, total and high-density lipoprotein (HDL) cholesterol, triglycerides, leptin, adiponectin, and c-reactive protein (CRP).

## 2. Materials and Methods

### 2.1. Overview

DUET was a single-blinded, 2-arm randomized controlled trial (RCT) that tested a 6-month web-based lifestyle intervention against a waitlist control among 56 dyads. Each dyad comprised a survivor of an obesity-related cancer, and their chosen partner, both of whom had obesity or overweight, were insufficiently active, and consumed suboptimal diets. This trial was approved by the University of Alabama at Birmingham (UAB) Institutional Review Board (300003882) and registered within ClinicalTrials.gov (NCT04132219). Detailed methods for DUET were published upon attainment of the accrual target of 56 dyads which ensued over a 9-month period ranging from October 2020 to July 2021 [11]; these methods are briefly summarized below.

### 2.2. Participants: Recruitment, Screening, Consent, and Randomization

Study invitations were distributed to adult survivors of localized renal cancer and loco-regional ovarian, colorectal, prostatic, endometrial, and female breast cancers (obesity-related cancers with 5-year survival rates >70%) identified from the UAB cancer registry, and listings of individuals expressing previous interest in lifestyle RCTs. Additionally, the Love Research Army (https://drsusanloveresearch.org/love-research-army) (accessed on 2 March 2023 initiated a series of email “blasts” to its members, and a recruitment website was established. Study staff provided telephone follow-up on mailings and contacts by placing up to six calls at various days and times. The study was explained, and interested survivors were screened for eligibility. Inclusion criteria were: (1) BMI ≥25 kg/m^2^; (2) moderate-to-vigorous physical activity (MVPA) < 150 min/week; (3) English speaking and writing; (4) educational attainment ≥5th grade; and (5) daily internet use and mobile phone ownership. Exclusion criteria were: (1) adhering to modified diets or enrolled in structured diet or exercise programs; (2) recent physician’s advice to limit PA and/or health issues precluding unsupervised PA or weight loss; and (3) residence in an assisted-nursing facility. Once eligibility was established and cancer case status (type and date of diagnosis) was verified by treating physicians of self-referrals, the survivor was asked to identify a partner with whom they interacted in person on at least a biweekly basis. Partners had identical inclusion/exclusion criteria (cancer survivorship was optional).

Telephone or Zoom calls were scheduled to review the study and acquire signed consent electronically (Adobe Sign^®^, San Jose, CA, USA). Participants completed baseline assessments, and their addresses were used to derive rural-urban commuting area codes (RUCA) as well as to estimate the distance between dyad members using Google Maps (https://www.google.com/maps) (accessed on 2 March 2023) since rural-urban status and proximity of dyad pairs could potentially affect access to healthy food procurement and exercise opportunities and support [9,12,13]. Dyads were randomly and evenly assigned to the DUET intervention or waitlist control using a permuted block design (block size = 4).

### 2.3. DUET Intervention

The DUET web-based intervention was adapted from two previously established programs: (1) the tailored mail-based, dyadic DAMES intervention, which was expanded to meet the needs of cancer survivors beyond those with just post-menopausal breast cancer and for partners beyond just biological daughters [9]; and (2) SurvivorSHINE, a web-based diet and exercise program for cancer survivors [14,15]. Like both of these interventions, DUET was theoretically grounded on Social Cognitive Theory (SCT) and emphasized skills training, modeling, incremental goal setting (with reinforcement), overcoming barriers, and self-monitoring (through the incorporation of new technologies, i.e., Fitbits and Aria Scales) [7]. Concepts from Interdependence Theory [16] and the Theory of Communal Coping [17] also guided the dyads’ commitment to relationship quality and the development of mutual goals to promote the adoption and maintenance of health behaviors and the provision and request for social support.

Upon randomization, one dyad member was mailed a box of supplies that included two sets of Portion Doctor ^®^ tableware, two Fitbits (Inspire^®^), two Aria 2^®^ digital scales, and two sets of instructions to connect to MyFitnessPal^®^ to automate weight and exercise tracking and provide additional reinforcement and support. Fitbit accounts also were linked to the password-protected, interactive DUET website, which formed the central core of the DUET intervention. Here each dyad member received tailored guidance over 24 weeks based on World Cancer Research Fund—American Institute of Cancer Research (WCRF-AICR) guidelines [18]. Thus, each dyad member was encouraged to set incremental goals that would eventually lead over the course of the 6-month intervention to exercising (including aerobic, resistance, flexibility, and balance) at least 150 min a week and adhering to a plant-based diet that included ample amounts of whole grains, vegetables and fruit (V and F), and limited amounts of red and processed meats, sugar, and refined (fast) food, while promoting a loss of roughly 0.5 kg per week. The website was designed with the following key features: (1) My Profile; (2) Topical Content; (3) Tip of the Day; (4) Sessions; (5) Tools; (6) News You Can Use; and (7) Support. Participants initially logged in to “My Profile” to enter age, height, gender, and current data on night-time snacking and intakes of V and F, whole and processed grains, red and processed meats, added sugars, supplement use, and alcohol. Survivors were prompted for data on cancer type, treatment, and coping style (Fighting Spirit or Fatalist) [19], which were used to provide tailored feedback, e.g., graphical displays with motivational messaging on overcoming treatment-related barriers (such as intolerance of high fiber V and Fs among survivors of colorectal cancer treated with a colostomy or urinary incontinence among survivors of prostate cancer), and calorie budgets to promote a loss of 0.5 kg w^−1^ [20]. Additionally, discrete tabs were provided to facilely reference topical information on healthy weight, healthy eating, and exercise. Daily tips for weight management, diet, and exercise were continually refreshed over the 6-month intervention as a means to enhance engagement. Furthermore, 24 weekly interactive sessions averaging 15 min in length were created using Articulate Storyline software (Articulate Global, LLC, New York, NY, USA) to guide participants through topics such as portion control, grocery shopping and food preparation, and various forms of exercise (aerobic, resistance, balance, and flexibility). A variety of tools also were provided on goal setting, customized meal plans, recipes, grocery lists, exercise guides, etc., in formats that could be downloaded and printed off. A tab entitled “News You Can Use” provided “take-away” summaries of recently released news stories and research pertaining to diet and exercise for cancer control. Finally, the webpage offered tips, such as active listening, to enhance dyad-based support. To enhance engagement with the website, Short Message System (SMS) text messages were issued thrice weekly. On each Monday of the 24-week intervention, dyads received a “push” message with a direct website link to the newly-released weekly session. On Wednesdays, dyads received a support message to reinforce the weekly content, and on Fridays, a “call-to-action” inquired about progress towards incremental goals.

### 2.4. Waitlist Control

Waitlisted dyads received all DUET resources and programming once 6-month follow-up data were collected.

### 2.5. Measures

Because DUET was implemented during COVID-19, several measures were adapted for remote assessment and were captured via Zoom^®^ (San Jose, CA, USA); validation study results were published previously [21].

#### 2.5.1. Anthropometric Measures (Captured at Baseline and 6 Months)

Body Weight (Primary Outcome): The participant stepped on the scale without shoes in light clothing while the process was captured via Zoom (focusing on the scale display to make sure the scale was “zeroed-out” prior to weighing). Weight was measured twice, with both values averaged for analyses.Waist circumference: This was measured with unmarked ribbons to reduce bias [22]; two sets of ribbons were mailed to each dyad for assessments. Participants bared their midriffs on Zoom and were instructed to place one end of the ribbon on the umbilicus. Partners then wrapped the ribbon evenly around the waist. Participants rotated in front of the camera, and assessors ensured the ribbon was parallel to the floor and snug against the skin. Upon exhale, partners used a felt-tip marker to mark the point of overlap. The process was repeated with the 2nd ribbon. Both ribbons were returned to the study office, where they were measured by staff, and the average was used for analyses.

#### 2.5.2. Dietary Intake (Captured at Baseline and 6 Months)

Two 24-h dietary recalls of a non-consecutive weekday and weekend day were conducted via telephone by a registered dietitian using the Automated Self-Administered (ASA-24) dietary assessment tool (https://epi.grants.cancer.gov/asa24) (accessed on 2 March 2023) at baseline and 6-months. Averaged intakes were obtained for calories, and diet quality was assessed using the Healthy Eating Index (HEI)-2015 [23].

#### 2.5.3. Physical Activity and Sleep (Captured at Baseline and 6 Months)

Objective PA data were captured using Actigraphs^®^ (Fort Walton, FL, USA) with instructions to wear the device at waist level on a provided belt during waking hours and to move the device to a provided wristband upon retiring to sleep. This procedure was followed for 7 days and accompanied by a written log. Minutes of MVPA were then downloaded and processed with similar methods used previously [24]. The Godin Leisure Time Exercise Questionnaire (GLTEQ) was administered online, giving excellent reliability and validity among cancer survivors [25].

#### 2.5.4. Physical Performance (Captured at Baseline and 6 Months)

Several physical performance measures were adapted for remote delivery and validated [21]. Dyads were mailed soccer cones, measuring tapes, 8′ lengths of cord, and stickers before assessments to perform measures. Trained assessors recorded images of testing via Zoom and then replayed them to capture accurate times and observations. Once data were entered into databases, videos were erased. Details of the remote assessment of the 30-s chair stand, 8′ get-up-and-go, sit-and-reach, back scratch, 2-min step test, and balance testing are reported by Pekmezi et al. [11] and Hoenemeyer et al. [21]

#### 2.5.5. Circulating Biomarkers (Captured at Baseline and 6 Months)

Participants received print and video instruction (https://youtu.be/lBPLS4PoHv4) (accessed on 2 March 2023) to self-collect 5 dried blood spots (DBS) on a designated card. These were dried for >4 h at room temperature, then inserted into a foil pouch with desiccant and frozen (0 F° or below) until analyzed. DBS eluents were batch-tested against known standards for insulin, glucose, leptin, adiponectin, high-density lipoprotein (HDL), and total cholesterol, triglycerides, and c-reactive protein (CRP) at the University of Washington as described previously [26]. To assure the validity of DBS assays in the current sample and data presented in this report, assays were performed using traditional multiplex methods on sera collected via phlebotomy from 36 participants at baseline and compared to a matched analysis of assays performed on DBS samples collected at the same time. Coefficients (R^2^) generated by ordinal logistic regression indicated strong correlations and were as follows: glucose = 0.981; CRP = 0.979; triglycerides = 0.979; total cholesterol = 0.963; leptin = 0.919; HDL = 0.899; adiponectin = 0.799; and insulin = 0.700. Values are expressed in plasma equivalent terms.

#### 2.5.6. Online Surveys

Online surveys were administered via REDCap^®^ (https://projectredcap.org) (accessed on 2 March 2023) at baseline, 3- and 6-months, though demographic information was only collected at baseline.

Comorbidity/Symptoms: The Older Americans Resources and Services (OARS) Comorbidity Index (modified version) assessed the number of medical conditions and functional impact by ascertaining either an affirmative or negative response to 21 unique medical conditions and 22 symptoms [27].Quality of Life (QOL): QOL was measured by the PROMIS global QOL to assess physical, mental, and social health domains, as well as usual activities, fatigue, and pain [28].Self-Efficacy is a central construct of SCT and is domain specific. Instruments with good internal consistency (α = 0.70–0.95) were selected to assess self-efficacy for dietary weight management [29] and exercise [30].Social Support: Validated 5-point scales by Sallis et al. [31] assessed social support for exercise and dietary change.Barriers: Thirty-one common barriers were assessed to a diet low in fat and sugar, increased V and Fs and whole grains, and exercise (e.g., “low calorie foods don’t taste good,” “I don’t know how to cook or prepare low calorie foods,” “I am too busy to (…follow a low calorie diet…exercise),” ”I don’t enjoy exercise, etc.) [32,33,34].Demographics: Self-reported data were collected on height, race/ethnicity, age, educational and marital status, current smoking status, income range, and relationship with the dyadic partner (i.e., spouse, child, parent, sibling, friend, or other). Because eHealth literacy also could serve as a potential moderator, three items from the eHEALS eHealth Literacy scale, [35], i.e., “I know how to use the Internet to answer my health questions; ”I know how to use the health information I find on the Internet to help me,” and “I have the skills I need to evaluate the health resources I find on the Internet” (5-item Likert scale ranging from strongly agree to disagree) were included in the baseline online survey. In addition, given substantial data showing that depression is a strong moderator of weight loss interventions [36], the Center for Epidemiologic Studies of Depression (CES-D; Boston short form; 20 items; yes-no format), was administered at baseline [37].

#### 2.5.7. Safety

All participants were encouraged to call a toll-free study number to report any adverse events. In addition, changes in health status were systematically ascertained in both study arms at 3 and 6 months. Events considered permanently disabling, life-threatening, or resulting in overnight hospitalization were deemed “serious,” with attribution to the intervention explored further.

### 2.6. Statistical Considerations

While accrual, retention, and safety form the basis of this feasibility trial, between-arm differences (i.e., differences between the intervention group and the waitlist control group) in weight loss (primary outcome) from baseline to 6 months were formally tested. Power calculations were performed using nQuery (version 8.5; GraphPad Software DBA Statistical Solutions, San Diego, CA, USA). These calculations assumed a standard deviation of 4.6 kg for the mean weight loss, as presented in our DUET protocol paper [11]. Assuming a sample size of 25 dyads/arm, a common standard deviation of 4.6 kg, a two-sided two-group *t*-test, and a significance level of 5%, there was >80% power to detect between-arm differences in weight loss of −3.72 kg or greater.

To determine whether important demographic and clinical characteristics of the sample were evenly distributed between the two study arms, the chi-square test (or Fisher’s exact test if the assumptions for the chi-square were not valid) for categorical study variables and the two-group *t*-test for continuous study variables were used. Distributions of continuous study variables were examined using stem-and-leaf, box, and normal probability plots and the Kolmogorov–Smirnov test; variables deviating from normal distribution were log_10_ transformed prior to analysis. All analyses were performed using SAS software (version 9.4; SAS Institute, Inc., Cary, NC, USA).

Arm differences in weight loss were assessed using an intent-to-treat approach. General linear mixed models, in particular, mixed model repeated measures analyses, were used to test for between-arm differences (two study arms), within-arm differences (three time points), and the arm-by-time point interaction simultaneously. These analyses were performed using PROC MIXED of SAS. This method accounts for the repeated measurements as well as the covariance between survivors, partners, or dyad members. A compound symmetry covariance matrix was assumed. This method provides tests of statistical significance (Type 3 tests which produce an F value and a *p*-value) for the between-arm effect, within-arm effect, and the interaction effect. When any of these effects were statistically significant, the Tukey–Kramer multiple comparisons test (performed using PROC MIXED of SAS) was used to determine which specific pairs of means for that effect were significantly different and also identified the time points at which those differences occurred. Such testing was helpful in comparing the multiple groups of survivors, partners, and dyads over two time points (when most outcomes were assessed) as well as three points (for survey data). Analyses were performed separately for survivors, partners, and combined dyads (thus, three sets of analyses were performed). Post-randomization exclusions (i.e., the three participants who either received gastrointestinal surgery or developed a cancer recurrence within 2 weeks of randomization) were omitted from 3- and 6-month analyses. Otherwise, all available data were used, though if a dyad member dropped out, data from that dyad were excluded from the dyadic analysis.

Analyses of secondary continuous outcomes were performed using general linear mixed models, as described in the previous paragraph for the primary outcome. These analyses were again followed by the use of the Tukey–Kramer multiple comparisons tests (for post hoc testing).

## 3. Results

### 3.1. Study Sample, Retention, and Safety

Sample characteristics are shown in Table 1. Overall, participants were diverse in terms of race, age, and geography (Alabama, Illinois, Mississippi, North Carolina, and Tennessee). Most were female, urban dwellers, and non-smokers, and roughly half reported being college graduates and currently employed with annual incomes >USD 50,000. Mean levels of V and F intake and PA were low as compared to the guidelines [5,6,18], while the average BMI was in the obese range, and participants reported an average of three other health conditions in addition to their cancer diagnosis. Survivors tended to be “long-term” (i.e., having diagnoses more than 5 years out), with most reporting early-stage cancers, of which a high proportion were breast cancer. A small number of previous cancer diagnoses (four breast, two gynecologic, and one testicular) were reported among partners. Given the high proportion of breast cancer survivors with spousal partners, survivors were significantly more likely to gender identify as female, while partners were more likely to report as male (*p*-values < 0.05). There were no statistically significant differences detected between the intervention vs. the waitlist control arms for any of the characteristics collected.

The CONSORT diagram (Figure 1) shows an enrollment rate of ~5.5% (*n* = 61/*n* = 1114). Of the 112 participants enrolled, three exclusions occurred within two weeks after randomization within the waitlisted arm (one survivor developed a cancer recurrence, another received emergency gastrointestinal surgery, and one partner received bariatric surgery), all of which were discontinued from the study and analysis, since all of these conditions affect the primary outcome (weight status). Additionally, three waitlisted partners were lost to follow-up (two of three dropped out when their survivor did so). Thus, the retention rate was 89% in the waitlisted arm and 100% in the intervention arm; the difference was statistically significant (*p* = 0.027).

Adverse events totaled 14 and 16 in waitlist and intervention arms, respectively. All events were non-attributable, and all except four were non-serious (two cancer recurrences, one myocardial infarction, and one acute cholecystitis), with no differences in events noted between the waitlist control and intervention and the intervention arms.

### 3.2. Changes in Adiposity

Significant weight loss occurred in both study arms (Table 2), though the magnitude of weight loss was significantly larger in survivors, partners, and dyads randomized within the DUET intervention arm. Dyads assigned to the DUET intervention lost significantly more weight (an average of 2.8 kg or 3.2% of their body weight) as compared to dyads that were waitlisted (who lost an average of 1.1 kg or 1.2% of their body weight). Findings related to waist circumference paralleled the results for weight loss, but differences between the two study arms did not reach statistical significance.

### 3.3. Changes in Dietary Intake and Physical Activity

Both study arms also significantly reduced their caloric intakes, with reductions being particularly notable among partners and dyads within the DUET intervention arm. However, calorie intake was significantly less among survivors assigned to the intervention arm than those randomized to the waitlist control (Table 2). While values for diet quality increased among intervention participants as compared to decreasing values among controls over the study period, these differences did not achieve statistical significance. Both study arms also showed significant increases in MVPA assessed either via self-report or accelerometry over the study period, though increases among survivors within the DUET study arm were of greater magnitude. That being said, differences between study arms did not reach statistical significance.

### 3.4. Changes in Physical Performance

As shown in Table 3, both study arms experienced significant improvements in several indices of physical performance (i.e., 30-s chair stand, 8′ get-up-and-go, sit-and-reach, and 2-min step test) over the study period, with DUET intervention arm

Survivors showed improvements of greater magnitude for all four tests and DUET dyads in 3-out-of-4 tests (i.e., all except the 2-min step test). DUET partners also showed notable improvements in the 30-s chair stand. However, in comparing improvements in the two study arms over time, significant differences were only detected for the flexibility measure, i.e., the sit-and-reach among survivors and dyads.

### 3.5. Changes in Circulating Biomarkers

As shown in Table 4, both study arms experienced significant decreases in circulating glucose, and while decreases were particularly noteworthy among DUET-assigned partners and dyads and among waitlisted survivors, these beneficial effects did not differ in statistical significance between study arms. Significant decreases over time also were observed among partners and dyads in both study arms for total cholesterol, as well as HDL cholesterol among all three subgroups (i.e., survivors, partners, and dyads). The effects on HDL cholesterol were particularly notable among dyad members of both study arms, with DUET dyads experiencing significantly greater decreases in HDL cholesterol than waitlisted dyads. Similarly, levels of CRP also decreased significantly over time among survivors and dyads in both study arms, and while these differences were particularly notable among survivors within the DUET intervention arm, no statistically significant differences were noted when the waitlist vs. the intervention arm were compared. Data on the adipokines, leptin, and adiponectin were less consistent, though significantly higher increases in leptin were observed among partners in the DUET intervention than among the waitlist control. No differences were detected for circulating levels of insulin or triglycerides.

### 3.6. Changes in Patient-Reported Outcomes

Significant improvements in physical QOL were observed over time among survivors of both study arms, though differences were not observed among other subgroups and also not for mental QOL. Further, no differences between the DUET intervention arm vs. the waitlist control were detected (Appendix A). While social support and self-efficacy for both diet and exercise increased over the 6-month period for intervention participants compared to decreasing levels among controls, these differences did not achieve statistical significance. In contrast, barriers decreased significantly over time in both study groups, with survivors and dyads reporting significantly fewer barriers to pursuing a low-calorie diet and partners and dyads reporting significantly fewer barriers toward exercise. While no statistically significant differences were identified between arms, *p*-values for time-by-arm interactions approached significance (e.g., *p* = 0.051).

### 3.7. Model Dyadic Terms

Of note, the models generated for DUET uncovered several significant dyadic terms, suggesting that the relationship established between the survivor and their partner appeared important for influencing effects on body weight (*p* = 0.009), waist circumference (*p* = 0.023), diet quality (*p* = 0.009), subjectively- and objectively-assessed PA (*p*’s < 0.001), sleep efficiency (*p* < 0.001), most physical performance tests (except the sit-and-reach) (*p*’s < 0.007), HDL cholesterol (*p* = 0.038), CRP (*p* = 0.002) and mental health (PROMIS; *p* < 0.001). Trends also were noted for caloric intake (*p* = 0.083), diet self-efficacy (*p* = 0.089), sleep fragmentation index (*p* = 0.0504), adiponectin (*p* = 0.095), and total cholesterol (*p* = 0.076).

## 4. Discussion

The DUET diet and exercise intervention was found to be feasible and resulted in significant weight loss among cancer survivors and their chosen partners. The 3.2% loss in body weight was not only statistically significant as compared to the 1.2% weight loss among controls but also is considered clinically significant and of the magnitude shown to exert favorable effects on glucose control and blood lipids by the American Heart Association, American College of Cardiology and The Obesity Society guidelines panel for the management of obesity and overweight [36]. This intervention is one of a handful of dyadic-based lifestyle interventions among cancer survivors [9,10,38,39,40] and among the few that promote change in multiple behaviors. Additionally, it is the only one that has employed a web-based platform. Moreover, while evidence is less consistent across both dyad members and as compared to the waitlist control, the DUET intervention also was associated with favorable effects on waist circumference, caloric restriction, self-reported and objective PA, as well as physical performance and blood glucose and CRP. While the relatively modest sample size may have limited power to detect differences in self-efficacy and social support, the intervention appeared to decrease the number of barriers affecting adherence to a calorically-restricted diet or increased PA. Thus, the theoretical concepts of SCT on which the DUET intervention was framed appear supported by these data and should be preserved in future trials. [7,11]

DUET achieved these favorable effects with minimal touch and excellent retention and safety; hence, results support future web-based interventions. Heretofore, variable success has been reported for multi-behavior, web-based interventions among cancer survivors. Bantum et al. [41] evaluated a comprehensive, 6-week, web-based symptom management program (including PA, weight management, and a healthful, plant-based diet) among 352 adult survivors of various cancers in Hawaii. The “Surviving and Thriving” intervention resulted in significant increases in moderate PA as compared to waitlisted controls but did not show concomitant increases in V and F intake (weight status was unreported). Kanera et al. [42] reported similar findings with a 6-month, web-based program (“Cancer Aftercare Guide”) among 462 Dutch survivors of mixed cancers and again found significant increases in self-reported moderate PA (+74.7 min/week) in the intervention arm but failed to detect significant increases in V and F intake in controlled analyses. By achieving improvements in calorie control and evidence of improvement in both self-reported and objectively-measured PA, DUET contributes to the unfolding science related to multi-behavior, web-based interventions and also demonstrates an impact on weight status. DUET is the first web-based intervention that reduced obesity and decreased caloric intake among cancer survivors, though, like the other studies also failed to detect significant differences in diet quality (albeit the sample size was six-to-nine-fold smaller and likely precluded our ability to detect between-arm differences).

DUET also promoted improvements in levels of CRP and glucose that have been observed in other more intensive weight loss interventions among cancer survivors, [43,44] though few differences were detected in other biomarkers typically associated with weight loss, i.e., leptin, adiponectin, and insulin [45,46]. Curiously, some participant groups (e.g., partners assigned to the intervention) experienced increases in leptin rather than decreases [46] and decreases in HDL cholesterol despite increased PA. Seasonal variation in circulating lipids may explain this latter finding since most participants were accrued during the summer when HDL cholesterol peaks and then descends towards its winter time nadir (corresponding to a 6-month follow-up) [47].

Unlike our previous Reach-out to ENhancE Wellness among older cancer survivors (RENEW) RCT that found significant improvements in both physical and mental QOL with a home-based diet and exercise weight management intervention among 641 breast, prostate, and colorectal cancer survivors [48], the current RCT only detected significant improvements in physical QOL, which it observed in both study arms. A probable explanation for this discrepancy was the smaller sample size, as the DAMES study [9], which also has a more modest sample size, likewise was unable to detect changes in this outcome.

An innovation of DUET was the expansion of dyadic composition beyond the family unit. This expansion increased the intervention reach and also likely enhanced DUET’s accrual (56 dyads in 9 months) versus Healthy Moves (22 dyads over 15 months) and DAMES (68 dyads over 2 years). While spouses and children still comprised roughly two-thirds of chosen partners, other family members, friends, and neighbors comprised the remainder. Furthermore, the significance of the dyadic term for most outcomes suggested that the synergy of the dyads was still strong, despite eligibility not being contingent on family relatedness. Yet, the inability to ascertain an eligible and willing partner served as an enrollment barrier. With social isolation and a lack of companionship reported at levels topping 20% in Western countries [49], and rates of living alone nearing 30% [50], there is an obvious need to explore effective “match-making” where cancer survivors could be paired with others based on factors that have the potential to create the synergy observed with naturally-occurring partnerships.

The DUET trial had several strengths. It formally tested a theoretically-grounded intervention using a randomized controlled design and validated, rigorous measures that were captured by trained staff blinded to study condition. Moreover, the study sample was diverse in geographic range and age, and the proportion of non-Hispanic Whites was representative of the U.S. population (62.5% vs. 59.3%) [50]. In addition, retention was excellent in both the intervention and waitlisted arms over the 6-month study period—exceeding the 70% benchmark that characterizes a tier-1 trial [51]; however, drop-out was significantly higher in the waitlisted arm. While differential drop-out can result in bias, this threat is considered minimal given the relatively small numbers (*n* = 6). Like many feasibility studies, DUET had a relatively modest sample size which may have undermined the statistical power to detect differences. Moreover, because of the focus on feasibility, statistical comparisons, of which there were many, were uncontrolled for multiple testing and may have uncovered spurious findings. The trial also did not assess whether weight loss and behavior changes were maintained long-term. Another weakness was the relatively low enrollment rate, i.e., uptake, especially among survivors of cancers other than breast, which may limit the generalizability of findings. However, the preponderance of breast cancer survivor participation in the current study is a phenomenon that has been reported commonly in traditionally delivered clinical interventions [52], as well as those that are digital [53].

## 5. Conclusions

DUET represents a pioneering effort in scalable, multi-behavior weight management interventions to promote cancer prevention and control. DUET not only demonstrated feasibility, safety, and excellent retention, it also promoted significant weight loss via caloric restriction and increased PA. Moreover, it exerted favorable effects on physical performance and markers of glucose metabolism and inflammation. Future studies are needed which are larger in size, scope, and duration and which assess longer-term effects of buddy-system interventions—interventions that include family members but that also extend cancer control to friends and neighbors.

## Figures and Tables

**Figure 1 cancers-15-01577-f001:**
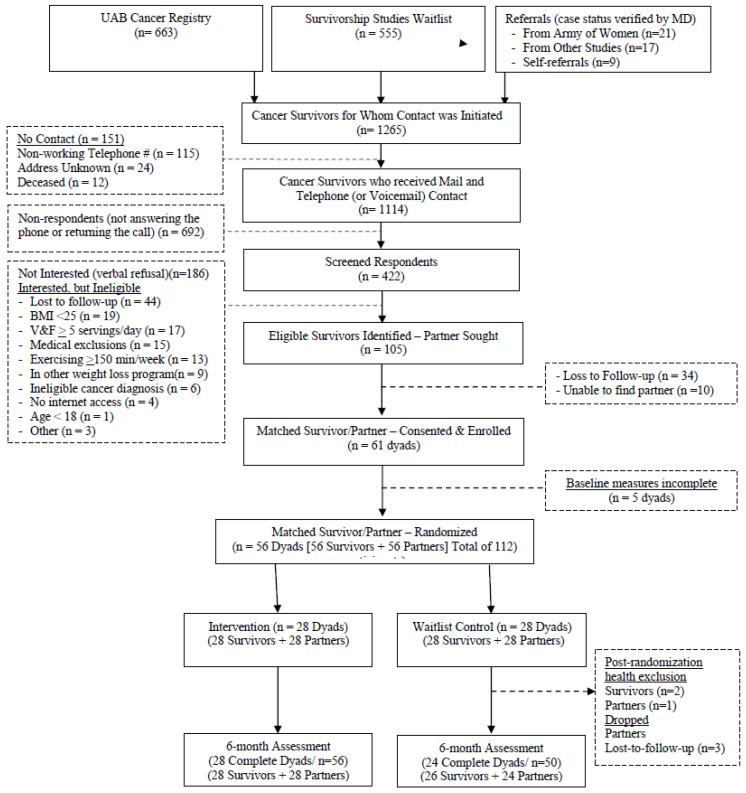
DUET CONSORT diagram.

**Table 1 cancers-15-01577-t001:** Characteristics of DUET study participants overall and by arm *.

Characteristic	Overall (*n* = 112)	Waitlist (*n* = 56)	Intervention (*n* = 56)
Cancer Diagnoses among Survivors (*n*/%) - Breast - Other †	45/56 (80.3%) 11/56 (19.7%)	22/28 (78.6%) 6/28 (21.4%)	23/28 (82.1%) 5/28 (17.9%)
Cancer Stage among Survivors (*n*/%) - I - II - III - Unknown	23/56 (41.1%) 6/56 (10.7%) 1/56 (1.8%) 26/56 (46.4%)	11/28 (39.3%) 3/28 (10.7%) 0/28 (0%) 14/28 (50%)	12/28 (42.9%) 3/28 (10.7%) 1/28 (3.6%) 12/28 (42.9%)
Months elapsed since diagnosis: Mean (sd)/range	67.3 (72.0)/0–303	63.1 (62.7)/12–206	71.0 (80.4)/10–303
Partner’s relationship to survivor, miles between them - Spouse - Sibling - Child - Friend - Other (parent, son-in-law, neighbor) Miles separating dyad members: Mean (sd)/range	23/56 (41.1%) 7/56 (12.5%) 6/56 (10.7%) 17/56 (30.4%) 3/56 (5.3%) 7.1 (12.4)/0–56	12/28 (42.9%) 5/28 (17.8%) 3/28 (10.7%) 8/28 (28.6%) 0/28 (0%) 5.7 (7.5)/0–31	11/28 (39.3%) 2/28 (7.2%) 3/28 (10.7%) 9/28 (32.1%) 3/28 (10.7%) 8.5 (16.0)/0–56
Female gender (*n*/%)	86 (76.8%)	42/56 (75.0%)	44/56 (78.6%)
Age (years): Mean (sd)/range	58.4 (12.7)/23–81	58.7 (13.0)/24–81	58.1 (12.5)/23–78
Race (*n*/%) - Non-Hispanic White - Non-Hispanic Black - Other Race or Ethnicity ‡	70 (62.5%) 41 (36.6%) 1 (0.9%)	34 (60.7%) 22 (39.3%) 0 (0%)	36 (64.3%) 19 (33.9%) 1 (1.8%)
Educational Status (*n*/%) - High School Degree or Less - Some College/Junior College/Trade School - College Degree/Post-Grad - Refused/Unknown	16 (14.3%) 34 (30.4%) 59 (53.5%) 2 (1.8%)	8 (14.3%) 14 (25.0%) 32 (57.1%) 2 (3.6%)	8 (14.3%) 20 (35.7%) 28 (50.0%) 0 (0%)
eHealth Literacy: Mean (sd)/range ¶	3.93 (0.77)/1.63–5.0	4.06 (0.78) 1.63–5.0	3.93 (0.77)/1.63–5.0
Income (*n*/%) - Less than USD50k/year - USD50k/year or more - Refused/Unknown	18 (16.1%) 48 (42.8%) 46 (41.1%)	7 (12.5%) 28 (50.0%) 21 (37.5%)	11 (19.6%) 20 (35.7%) 25 (44.6%)
Employment (*n*/%) - Employed - Retired - Other (student, disabled)	62 (55.4%) 36 (32.1%) 14 (12.5%)	32 (57.1%) 17 (30.4%) 7 (12.5%)	30 (53.6%) 19 (33.9%) 7 (12.5%)
Rural Residence (*n*/%)	9 (8.0%)	6 (10.7%)	3 (5.4%)
BMI: Mean (sd)/range	32.4 (2.75) 25–51	33.3 (6.8) 25–51	31.4 (4.9) 25–45
Daily servings of Vand Fs: Mean (sd)/range	1.8 (1.1) 0.2–6.3	1.9 (1.2) 0.2–6.3	1.7 (1.0) 0.2–3.8
Weekly minutes of MVPA: Mean (sd)/range	43.8 (54.7) 0–280	43.7 (59.1) 0–225	43.8 (60.5) 0–280
Current Smoker (*n*/%)	5 (4.5%)	1 (1.8%)	4 (7.1%)
Number of Comorbidities: Mean (sd)/range	3.0 (2.6) 0–11	3.5 (2.7) 0–11	2.6 (2.4) 0–10
Depressive Symptoms (CESD): Mean (sd)/range §	3.2 (6.3) 0–24	3.9 (6.1) 0–24	3.9 (6.1) 0–24

* No statistically significant differences were observed between the intervention vs. the waitlist control arm. † Other cancer diagnoses among survivors were colorectal (*n* = 1), gynecologic (*n* = 2), renal (*n* = 2), and prostatic (*n* = 6). ‡ Hispanic ethnicity in combination with various racial groups. ¶ Potential range 1–5. § Potential range 0–24.

**Table 2 cancers-15-01577-t002:** Baseline to 6-m differences in adiposity, diet, and physical activity in waitlist vs. DUET intervention study arms.

	Waitlist Control (WL)		DUET Intervention		Significance (*p*-Values)		
	Baseline Mean (SD)	6M Mean (SD)	Baseline Mean (SD)	6M Mean (SD)	Between Arm	Within Arm	Time × Arm
**Weight (kg)** *(Primary Outcome)*							
Survivors Partners Dyads	88.0 (17.6) 96.7 (22.6) 92.4 (20.6)	86.9 (16.1) 95.3 (22.9) 91.3 (20.3)	86.8 (18.2) 87.6 (14.0) 87.2 (16.1)	83.8 (18.4) * 84.9 (15.6) * 84.4 (16.9) *	0.656 0.069 **0.044**	**0.001** **0.001** **<0.001**	0.090 0.280 **0.033**
Waist Circumference (cm)							
Survivors Partners Dyads	107.1 (17.7) 111.3 (15.3) 109.2 (16.7)	104.6 (14.3) 108.4 (16.4) * 106.6 (15.6) *	106.0 (11.7) 106.5 (9.5) 106.2 (10.6)	102.5 (13.4) * 103.3 (9.5) * 102.9 (11.5) *	0.667 0.144 0.128	**0.003** **<0.001** **<0.001**	0.579 0.594 0.339
Calorie Intake/day Survivors Partners Dyads	1645.9 (535.0) 1553.5 (483.7) 1596.9 (515.6)	1532.0 (568.9) 1467.2 (501.3) 1497.7 (542.5)	1400.0 (439.0) 1570.2 (498.3) 1485.1 (473.1)	1265.6 (305.1) 1378.5 (408.3) * 1321.0 (360.6) *	**0.027** 0.716 0.053	**0.046** **0.012** **0.001**	0.836 0.286 0.365
Healthy Eating Index Survivors Partners Dyads	51.8 (10.9) 55.5 (10.5) 54.0 (10.6)	50.8 (12.2) 54.6 (12.2) 52.6 (12.5)	53.9 (13.7) 52.2 (12.0) 53.1 (12.8)	58.8 (15.2) 53.9 (13.4) 56.4 (14.4)	0.092 0.416 0.396	0.360 0.874 0.589	0.163 0.587 0.130
MVPA Self-Report Survivors Partners Dyad	51.9 (61.6) 43.3 (61.8) 45.6 (61.4)	57.3 (61.9) 58.1 (75.4) 57.9 (68.6)	48.5 (67.8) 39.1 (52.9) 43.1 (60.4)	103.9(104.7) * 81.6(113.5) 92.7(108.8) *	0.448 0.299 0.099	**0.011** **0.025** **<0.001**	0.189 0.356 0.229
MVPA Accelerometry Survivors Partners Dyads	162.6 (172.4) 148.0 (169.0) 157.7 (171.3)	122.2 (190.2) 146.0 (278.4) 139.0 (236.8) †	108.5 (122.6) 146.4 (121.8) 127.5 (122.4)	150.3 (199.0) * 132.0 (271.4) 141.0 (236.4)	0.726 0.655 0.633	**0.047** **<0.001** **<0.001**	0.265 0.819 0.600

* Post-hoc analyses show significant improvements from baseline (*p* < 0.05); † Post-hoc analyses show significant declines from baseline (*p* < 0.05).

**Table 3 cancers-15-01577-t003:** Baseline to 6-m differences in physical performance in waitlist vs. DUET intervention study arms.

	Waitlist Control (WL)		DUET Intervention		Significance (*p*-Values)		
	Baseline Mean (SD)	6M Mean (SD)	Baseline Mean (SD)	6M Mean (SD)	Between Arm	Within Arm	Time × Arm
30-sec Chair Stand (reps) Survivors Partners Dyads	10.5 (2.7) 11.5 (3.0) 11.0 (2.9) *	12.2 (3.9) 14.0 (5.8) 13.1 (5.0)	10.3 (3.1) 11.0 (3.4) 10.7 (3.2)	13.1 (3.6) * 13.5 (3.3) * 13.3 (3.4) *	0.645 0.624 0.871	**<0.001** **<0.001** **<0.001**	0.301 0.995 0.380
8′ Get-up-and-Go (sec) Survivors Partners Dyads	8.1 (2.7) 7.4 (1.5) 7.8 (2.3)	7.6 (2.8) 7.3 (1.8) 7.6 (2.3)	7.7 (1.7) 7.6 (1.6) 7.6 (1.6)	6.9 (1.5) * 7.1 (2.0) 7.0 (1.8) *	0.290 0.980 0.107	**0.004** 0.084 **0.001**	0.470 0.304 0.211
Sit-and-Reach (cm) Survivors Partners Dyads	−0.2 (4.6) 0.3 (3.6) 0.1 (4.0)	−0.3 (4.7) 0.8 (3.6) 0.4 (4.1)	−0.8 (3.0) −0.5 (3.8) −0.6 (3.4)	0.6 (2.0) * 0.5 (2.8) 0.6 (2.4) *	0.889 0.583 0.573	**0.036** **0.025** **0.002**	**0.015** 0.520 **0.043**
Back Scratch (cm) Survivors Partners Dyads	−3.7 (3.5) −2.5 (3.4) −3.0 (3.4)	−3.7 (3.2) −2.3 (3.3) −2.9 (3.3)	−3.3 (3.5) −3.7 (3.6) −3.5 (3.6)	−2.9 (3.9) −3.3 (3.6) −3.1 (3.7)	0.627 0.260 0.511	0.187 0.237 0.104	0.217 0.699 0.147
2-min Step Test (reps) Survivors Partners Dyads	81.0 (22.9) 79.7 (20.8) 80.3 (22.0)	80.4 (23.0) 83.0 (22.5) 80.9 (22.6)	79.9 (23.1) 79.1 (24.2) 79.4 (23.4)	84.4 (23.5) * 87.3 (23.6) 85.8 (23.4)	0.806 0.766 0.442	0.396 **0.016** **0.037**	0.258 0.298 0.077

* Post-hoc analyses show significant improvements from baseline (*p* < 0.05).

**Table 4 cancers-15-01577-t004:** Baseline to 6-month differences in circulating biomarkers in waitlisted vs. DUET intervention study arms.

	Waitlist (WL) Control		DUET Intervention		Significance (*p*-Values)		
	Baseline Mean (SD)	6M Mean (SD)	Baseline Mean (SD)	6M Mean (SD)	Between Arm	Within Arm	Time × Arm
Glucose (mg/dL) Survivors Partners Dyads	87.8 (25.6) 88.2 (17.8) 88.3 (22.6)	62.6 (34.2) * 65.8 (44.3) 61.2 (38.0)	89.0 (23.1) 82.7 (21.6) 85.9 (22.4)	71.4 (23.1) 67.4 (34.5) * 69.4 (29.2) *	0.175 0.599 0.185	**0.002** **0.002** **<0.001**	0.188 0.393 0.148
Total Chol. (mg/dL) Survivors Partners Dyads	269.5 (69.6) 280.3 (58.2) 276.9 (64.4)	252.1 (34.0) 238.1 (42.1) 245.6 (39.2)	268.2 (50.7) 269.5 (59.4) 268.8 (54.6)	255.5 (44.8) 249.3 (38.4) 252.3 (41.3)	0.713 0.969 0.868	0.139 **0.005** **0.002**	0.791 0.303 0.377
HDL Chol. (mg/dL) Survivors Partners Dyads	74.2 (11.7) 72.3 (10.8) 73.5 (11.4)	68.1 (14.4) † 67.6 (12.7) 68.0 (13.8) †	68.4 (10.9) 69.0 (11.9) 68.7 (11.3)	63.2 (10.6) 61.2 (10.5) † 62.1 (10.5)†	0.130 0.119 **0.017**	**<0.001** **0.002** **<0.001**	0.593 0.405 0.977
CRP (mg/dL) Survivors Partners Dyads	2.0 (1.4) 3.2 (2.6) 2.7 (2.1)	1.7 (1.5) 2.7 (2.2) 2.3 (1.9)	4.5 (7.4) 2.7 (4.0) 3.6 (6.0)	3.0 (3.5) 2.3 (4.5) 2.6 (4.0) *	0.182 0.062 0.403	**0.041** 0.142 **0.025**	0.824 0.328 0.183
Leptin Survivors Partners Dyads	102.2 (71.6) 113.0 (76.7) 108.1 (75.0)	91.6 (50.2) 116.0 (79.3) 103.7 (68.6)	101.8 (65.1) 74.0 (51.0) 88.2 (59.7)	100.9 (86.5) 83.8 (74.8) 92.1 (80.1)	0.767 **0.017** 0.161	0.689 0.945 0.705	0.735 0.523 0.448
Adiponectin Survivors Partners Dyads	34.6 (23.1) 26.3 (9.9) 31.1 (18.8)	31.9 (15.2) 24.7 (9.4) 28.4 (13.4)	26.8 (20.8) 25.2 (13.6) 26.0 (17.5)	23.8 (11.1) 23.3 (12.6) 23.5 (11.7)	0.283 0.380 0.113	0.343 0.174 **0.043**	0.137 0.247 0.408

* Post-hoc analyses show significant improvements from baseline (*p* < 0.05); † Post-hoc analyses show significant declines from baseline (*p* < 0.05)

## Data Availability

The data presented in this study are available on request from the corresponding author.

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
