# Peer review of "Results of DUET: A Web-Based Weight Loss Randomized Controlled Feasibility Trial among Cancer Survivors and Their Chosen Partners"

_cancers, 2023, doi:10.3390/cancers15051577_

Round 1

Reviewer 1 Report

This is an interesting and well written paper. The novelty is appealing and shows promise for implementation. I think the family unit involvement is more significant than the authors do, but given the timing of the study during COVID, not surprising. I also wonder about the use of dried blood spots. Do the authors have any reliability and accuracy data to support the use of these measures? Some formatting is required on the Tables. Alignments of the first column information may have been altered with the line numbering?

Author Response

This is an interesting and well written paper. The novelty is appealing and shows promise for implementation. I think the family unit involvement is more significant than the authors do, but given the timing of the study during COVID, not surprising. I also wonder about the use of dried blood spots. Do the authors have any reliability and accuracy data to support the use of these measures? Some formatting is required on the Tables. Alignments of the first column information may have been altered with the line numbering? 

We appreciate the positive comments – thank you!  Previously, we cited the validity studies conducted at the University of Washington where our analyses were performed.  Now, we have added the following verbiage on the correlations between assays conducted on sera collected by phlebotomy on our study sample at baseline and assays using dried blood spots on the same sample: 

To assure the validity of DBS assays in the current sample and data presented in this report, assays were performed using traditional multiplex methods on sera collected via phlebotomy from 36 participants at baseline and compared to a matched analysis of assays performed on DBS samples collected at the same time. Coefficients (R2) generated by ordinal logistic regression indicated strong correlations and were as follows: glucose=0.981; CRP=0.979; triglycerides=0.979; total cholesterol=0.963; leptin=0.919; HDL=0.899; adiponectin=0.799; and insulin=0.700 (see lines 252-259 clean copy). 

In reviewing these data, we have chosen not to report IL-6 since the correlations was only 0.447. We have made the necessary formatting edits to our tables. 

Reviewer 2 Report

General comment: This article describes results of a feasibility trial that tested an intervention for obese cancer survivors and chosen partners in improving various outcomes related to obesity and physical activity. The topic is interesting, and the procedure and the methods seem robust. As a major limitation, the articles includes a very high number of outcomes and statistics. Given that the statistics and their differences are insufficiently described in the analysis section, the results and the discussion are hard to follow. Therefore, my main concerns are to clarify the analysis section, to streamline results regarding outcomes and statistics and to revise the discussion section accordingly.

Abstract:

-          Methods: Please include more info about the analyses here; outcomes should be presented here

-          Results: please give p-values for all results you mention here;

-          non-significant findings also need to be mentioned

Introduction:

-          Generally, the introduction reads well

-          However, the presentation of previous intervention studies and the current one is too technical in my opinion (e.g., information about p-values or detailed methodological information); such information should be shifted in the discussion to compare results to each other; here, I would rather be interested in the main efforts of and problems with previous research and why and how exactly the current study tries to take into account these issues

Overview

-          “…both of whom had obesity…” does that mean that also the chosen partner need to have overweight problems? If this is the case, a short rationale for this approach may be needed here

Participants

-          What was the rationale to derive RUCA and distance between dyad members? Did the participants know about this procedure? Please clarify

2.5 Measures

-          I agree that the measures are complex and need more description than questionnaires; nevertheless, I think that information may be streamlined; additional details may be shifted in the appendix

-          please clarify why the online surveys were conducted at 3- and 6-month follow-up while the rest of outcomes were assessed at 6-months follow-up only

-          there is a very high number of outcomes: have all of these been defined in the study protocol on clinicaltrials? if so, there needs to be a brief justification of the need of the analysis of so many outcomes

Statistical considerations

-          does the first paragraph explain the procedure of sample size calculation? Please clarify and re-formulate this passage

-          second paragraph: “between-arm differences were assessed with…”: which differences are you reffering to? Sample characteristics or outcomes? Please clarify

-          “Post-randomization exclusions were excluded from 3- and 6-month analyses.” I did not get this sentence, please re-formulate

-          Given that not all outcomes were assessed at 3- and 6-months-follow-up, it should be explicitly described which outcome was analyzed for which time point

-          The last paragraph remains unclear: what is meant with “similar approach as described” – which approach exactly did you use? Also, the analysis of categorical data with general linear mixed models may be even more complex than with continous variables and thus a more detailed description is needed

-          Did you apply any effect sizes or thresholds to assess practical relevance of findings? Please clarify

-          The approach of providing between arm, within arm, and time x arm testing results need more explanation here: what is the exact approach and rationale for each of these statistics and what are the differences in their interpretation regarding the research question; there needs to be more details here to be able to understand the results

-          I also miss details and rationale of the post hoc testings

Results:

-          Table 2: this table needs some re-formatting

-          “Mean levels of V&F intake and PA were low” – how do you come to this conclusion? Did you have any normative values?

-          The CESD is given in the table, but I could not find it in the methods

-          Paragraph related to CONSORT flow: “the between-arm difference was statistically significant (p=0.027).” what is the difference referring to? Please clarify

-          Flow chart:

o   What is meant with “…survivors for whom contact was assumed”? please clarify

o   What is the difference between non-respondents and those who “refused” among the screened respondents

o   It seems that one box in the low chart does not show all of its content

-          The results of tables 3 to 6 are hard to interpret given the insufficient description of the 3 statistics in the analysis section (see above)

-          The text outlining the results should be streamlined

-          I strongly suggest to streamline the whole result sections: it should be re-considered which outcomes are needed to be presented in the manuscript and which may be shifted in the appendix

-          Also, it might be re-considered if all of the statistics need to be described here and which of them may be provided in the appendix; up to now, it is not possible for the reader to get a clear message on the efficacy of the trial

Discussion

-          The first paragraph outlines the strengths of the study, but should present the main findings

-          Given that the results section including the various outcomes and statistics was hard to understand, it is also hard to follow the discussion; please re-vise accordingly

-          As with the introduction, the discussion is too technical in my opinion; it rather should put the results into a broader context

Author Response

General comment: This article describes results of a feasibility trial that tested an intervention for obese cancer survivors and chosen partners in improving various outcomes related to obesity and physical activity. The topic is interesting, and the procedure and the methods seem robust. As a major limitation, the articles includes a very high number of outcomes and statistics. Given that the statistics and their differences are insufficiently described in the analysis section, the results and the discussion are hard to follow. Therefore, my main concerns are to clarify the analysis section, to streamline results regarding outcomes and statistics and to revise the discussion section accordingly.  We appreciate the suggestions of the reviewer and have made several changes to our results and discussion.   Because of the stringent restrictions on the abstract word count (i.e., 200), it was not possible to address all of the suggestions made for that component.

Abstract:

-          Methods: Please include more info about the analyses here; outcomes should be presented here                                                While the abstract needed to be heavily edited to accommodate a phrase regarding statistical testing, this was accomplished.

-          Results: please give p-values for all results you mention here                                                                                                              We have inserted p-values for caloric intake in addition to body weight.

-          non-significant findings also need to be mentioned We apologize, but with the abstract capped at 200 words, this was not possible.

Introduction:

-          Generally, the introduction reads well Thank you.

-          However, the presentation of previous intervention studies and the current one is too technical in my opinion (e.g., information about p-values or detailed methodological information); such information should be shifted in the discussion to compare results to each other; here, I would rather be interested in the main efforts of and problems with previous research and why and how exactly the current study tries to take into account these issues                                                          Given that the other reviewers commended the introduction, we hesitate to make too many changes, particularly since this article is most likely to be read and cited by cancer researchers who have expertise in cancer control, behavioral science, or supportive care (a substantial and important target readership for CANCERS).  That being said, we have removed some detail, as well as the p-values in order to try to address the reviewer’s concern, and because the verbiage, “statistically significant” implies p-values that are <0.05.  We also have removed abbreviations to enhance readability.  We hope that these edits satisfy reviewer #2. 

Overview

-          “…both of whom had obesity…” does that mean that also the chosen partner need to have overweight problems? If this is the case, a short rationale for this approach may be needed here.

      Both the cancer survivor and the partner had a BMI of 25 and above.  We believe that the first paragraph provides the rationale that excess adiposity is a risk factor for both primary prevention and tertiary prevention of cancer; however, to reduce any confusion, we have inserted additional text reinforcing the point that DUET was an intervention aimed at both the cancer survivor (for the purposes of cancer control and tertiary prevention) and the partner (for primary cancer prevention).   

Participants

-          What was the rationale to derive RUCA and distance between dyad members? Did the participants know about this procedure? Please clarify                                                                                                                                                     RUCA codes were generated to characterize whether participants were residents of rural vs. urban environments since food procurement and exercise opportunities may vary accordingly.  The physical distance between dyad members was ascertained because opportunities to exercise together, as well as to procure healthy foods and prepare recipes may be affected by physical proximity.  To address the reviewer’s concern, we have added the phrase, “since rural-urban status and proximity of dyad pairs could potentially affect access to healthy food procurement and exercise opportunities and support,” and provide appropriate references,   To assure that randomization was effective in evenly distributing these potential effect modifiers, we ascertained them at baseline as key demographic variables, for which study participants well apprised.  Indeed, randomization was effective in distributing these factors evenly among study arms, thus eliminating any need for adjustment.

2.5 Measures

-          I agree that the measures are complex and need more description than questionnaires; nevertheless, I think that information may be streamlined; additional details may be shifted in the appendix.                                                                 Since this reviewer, as well as the other two reviewers are requesting more detail on some of the methods, we believe that moving details of the methods into the appendix would raise more issues than solve them. Instead, we carefully evaluated the methods that already were reported in detail in both our previously reported methods paper, as well as in our validation study and directed the reader to those sources for details.

-          please clarify why the online surveys were conducted at 3- and 6-month follow-up while the rest of outcomes were assessed at 6-months follow-up only.                                                                                                                                       Yes, the reviewer is correct, all assessment components were implemented at baseline and 6-month follow-up, whereas  online surveys also were conducted at 3-months (primarily to capture changes in the factors that are associated with uptake on the intervention and behavior change). 

-          there is a very high number of outcomes: have all of these been defined in the study protocol on clinicaltrials? if so, there needs to be a brief justification of the need of the analysis of so many outcomes.                                                                 As stated in the protocol, the primary outcomes for this pilot study relate to feasibility (i.e., recruitment, retention, acceptability and safety) and change in body weight, others are strictly secondary and were explored primarily to inform future trials.  Given that diet and exercise interventions are not only designed to impact targeted behaviors (diet and physical activity which are assessed via multiple means), but also are mediated through targeted behavioral constructs (e.g., self-efficacy, barriers, social support), and also can impact quality of life, physical functioning and biomarkers, there are several secondary outcomes that are commonly assessed in behavioral intervention trials.  Indeed, 79 outcomes are included in the protocol posted on ClinicalTrials (NCT04132219).  Verbiage is provided in multiple places within the text to make the distinction between primary and secondary outcomes (see lines 40, 102, 104, 210, 302, 337, 338, 375, and Table 2).

Statistical considerations (The following responses were prepared by biostatistician, Dr. Robert Oster, who was recently selected as a Fellow of the American Statistical Association and who performed the statistical analyses on the study)

-          does the first paragraph explain the procedure of sample size calculation? Please clarify and re-formulate this passage                 Yes, the first paragraph explains the procedure that we used for our sample size calculation. We reworded this section to improve clarity.

-          second paragraph: “between-arm differences were assessed with…”: which differences are you reffering to? Sample characteristics or outcomes? Please clarify.)                                                                                                                                     We assessed differences between the two study arms (intervention group and waitlist control group) for the sample characteristics (demographic and clinical). Differences for the outcomes were not assessed using these methods (instead, these differences were assessed using general linear mixed models using PROC MIXED of SAS, as described in the third paragraph). We have revised the wording accordingly.

-          “Post-randomization exclusions were excluded from 3- and 6-month analyses.” I did not get this sentence, please re-formulate                                                                                                                                                                                                   Data from the three individuals who developed a cancer recurrence or who underwent G.I. surgery (conditions that were likely to influence the primary endpoint of weight loss) within two weeks of randomization were not included in the analysis – see lines 332-334 and 371-375 clean copy for clarification. 

-          Given that not all outcomes were assessed at 3- and 6-months-follow-up, it should be explicitly described which outcome was analyzed for which time point.                                                                                                                                                                       Of all outcomes, only quality of life and factors associated with the behavioral intervention framework (self-efficacy, social support, and barriers) were assessed at three timepoints, instead of two.  Statistical methods however were identical (and the post-hoc testing addresses the reviewer’s concern about identifying differences by group and the timepoint at which specific differences were detected. 

-          The last paragraph remains unclear: what is meant with “similar approach as described” – which approach exactly did you use? Also, the analysis of categorical data with general linear mixed models may be even more complex than with continous variables and thus a more detailed description is needed                                                                                                                        In the last paragraph, we clarified the approach that we used for the analysis of the secondary continuous outcome. We also added more details for our analysis of categorical data. We used generalized linear mixed models for these analyses (using PROC GLIMMIX of SAS). We revised the wording accordingly.

-          Did you apply any effect sizes or thresholds to assess practical relevance of findings? Please clarify                                          Thank you for pointing-out this oversight.  A 3% weight loss, as observed among both survivors and partners assigned to the DUET intervention is associated with improved blood lipids and glucose control and thus is considered clinically significant as indicated by the practice guidelines issued by The Obesity Society, the American College of Cardiology and the American Heart Association Task Force.  This has been added to the first paragraph of the discussion and supported be the reference of Jensen et al. (2014).  (see lines 456-461 clean copy).  

-          The approach of providing between arm, within arm, and time x arm testing results need more explanation here: what is the exact approach and rationale for each of these statistics and what are the differences in their interpretation regarding the research question; there needs to be more details here to be able to understand the results                             In the third paragraph, we now provide more details for the approach that we used for the analysis of the primary outcome. We revised the wording accordingly.

-          I also miss details and rationale of the post hoc testings                                                                                                                          As per the reviewer’s suggestion and in the third paragraph, we now provide more details regarding our use of the post hoc testing. We revised the wording accordingly.

Results:

-          Table 2: this table needs some re-formatting.                                                                                                                                          We apologize, this table become disorganized upon upload, and we failed to notice.  We have made the necessary corrections. 

-          “Mean levels of V&F intake and PA were low” – how do you come to this conclusion? Did you have any normative values?                                                                                                                                                                                                 Levels were low as compared to the guidelines which recommend eating 5 servings of V&F/day and exercising 150 minutes/week.  We have clarified this statement and have added references for the guidelines (see line 347-354 clean copy)

-          The CESD is given in the table, but I could not find it in the methods                                                                                                  We apologize for this oversight, the CES-D and the rationale for collecting data related to depression have been added to the bullet point on demographics (lines 289-292 clean copy). 

-          Paragraph related to CONSORT flow: “the between-arm difference was statistically significant (p=0.027).” what is the difference referring to? Please clarify                                                                                                                                               The sentence has been edited to now read: Thus, the retention rate  was 89% in the waitlisted arm and 100% in the interventions arm; the difference was statistically significant (p=0.027).

 Flow chart:

o   What is meant with “…survivors for whom contact was assumed”? please clarify                                                                                To improve clarity, the box now reads Cancer Survivors who received Mail and Telephone (or Voicemail) Contact 

o   What is the difference between non-respondents and those who “refused” among the screened respondents:                                      Non-respondents are those who by all accounts, received the letter and voicemail and who chose never to pick-up the phone or return the message, whereas refusals verbalize that they do not want to participate.  Although, both terms are used commonly in mail- and telephone-based recruitment and are generally understood, we have sought to clarify the distinction by parenthetically adding the following: (not answering the phone or returning the call) to the Non-respondent box.  Additionally, we have replaced the word “Refused” with “Not Interested, (verbal refusal).”   

o   It seems that one box in the low chart does not show all of its content                                                                                                    We have corrected this issue

-          The results of tables 3 to 6 are hard to interpret given the insufficient description of the 3 statistics in the analysis section (see above)                                                                                                                                                                                  We have simplified the tables; moreover, we have moved the former Table 6 to the supplemental materials

-          The text outlining the results should be streamlined. I strongly suggest to streamline the whole result sections: it should be re-considered which outcomes are needed to be presented in the manuscript and which may be shifted in the appendix.                                                                                                                                                                                                     As per the reviewer’s suggestion, we have streamlined the results, using the appendix (supplementary Tables to describe outcomes that are more tangential).  We also rearranged the table of biomarkers to better correspond to the streamlined discussion.

-          Also, it might be re-considered if all of the statistics need to be described here and which of them may be provided in the appendix; up to now, it is not possible for the reader to get a clear message on the efficacy of the trial.                                                       We appreciate the reviewer’s suggestion and believe that the modifications to the section of Statistical Considerations now provides sufficient detail.

Discussion

-          The first paragraph outlines the strengths of the study, but should present the main findings                                                                       The first paragraph does not detail the strengths, but rather the key factors that “set this research apart” from other studies.  Many research articles are structured similarly; however, in order to be responsive to the reviewer’s suggestion, we have incorporated the main findings within this initial paragraph.

-          Given that the results section including the various outcomes and statistics was hard to understand, it is also hard to follow the discussion; please re-vise accordingly                                                                                                                                              As per the reviewer’s suggestion, we have revised the discussion to improve clarity.

-          As with the introduction, the discussion is too technical in my opinion; it rather should put the results into a broader context.                                                                                                                                                                                         Please see second response to the comment under the Introduction.

Reviewer 3 Report

The study addresses an important area of cancer research and survivorship care, your retention rate is commendable given the population and intervention. 

1. Table 5 shows subjects to largely have elevated total cholesterol levels as well as surprisingly high HDL levels. Any thoughts on why this population has such desirable HDL levels?

2. It is interesting that glucose levels declined so drastically from baseline (table 5). Was the diet a carbohydrate-restricted diet?

3. Could you better define diet-related and exercise related "barriers"? I was unable to understand the term "barriers" as used here.

4. The QoL and mental health neither improved nor worsened. Do you see this as an expected outcome or did you expect a difference? 

5. A suggestion: The tables are too wide with too much data distributed across rows. It makes the reading difficult, so I wonder if there is an alternate formatting option or perhaps leave some data out to help the relevant data be seen more easily/ 

Author Response

The study addresses an important area of cancer research and survivorship care, your retention rate is commendable given the population and intervention.  Thank you!

  1. Table 5 shows subjects to largely have elevated total cholesterol levels as well as surprisingly high HDL levels. Any thoughts on why this population has such desirable HDL levels? The reviewer brings up an interesting point; however, as noted, total cholesterol levels are elevated in this sample, and because HDL cholesterol contributes to total cholesterol, there is a well-established positive correlation between the two (Davis et al. Circulation, 62: 24-30, 1980).  Therefore, it is not as unusual as it appears and the Total cholesterol: HDL cholesterol ratio is well under the 4-5 range advocated by the American Heart Association.
  2. It is interesting that glucose levels declined so drastically from baseline (table 5). Was the diet a carbohydrate-restricted diet? The plant-based diet advocated by the World Cancer Research Fund – American Institute of Cancer Research and the American Cancer Society is not carbohydrate restricted, nor did we observe differences in macronutrient distribution in this study.  Instead, the drop in glucose is likely due to the reductions in weight, corroborating the report of the American College of Cardiology/American Heart Association Task Force and The Obesity Society’s Practice Guidelines that only 3% weight loss may be necessary to improve glucose control.  We now discuss this more thoroughly in the initial sentences of paragraph 1 of the discussion (see lines 456-561 clean copy).   
  3. Could you better define diet-related and exercise related "barriers"? I was unable to understand the term "barriers" as used here. Overcoming barriers to behavior change is a central concept of Social Cognitive Theory (Bandura 1986); thus, it is a key strategy used in many behavioral interventions.  For this study, we measured 31 common barriers to exercise and eating a healthful, low calorie, plant-based diet proposed by Sallis et al. 1987 and others.  While we do not have room to list all 31, we have expanded the examples listed in the methods to provide further clarity (see lines 276-279 clean copy). 
  4. The QoL and mental health neither improved nor worsened. Do you see this as an expected outcome or did you expect adifference? 

Actually, we had hoped for the differences observed in our previous RENEW trial.  To respond to the reviewer’s interest, we have added the following brief paragraph to the discussion (see lines 501-508)  just in case other readers have similar interests.  

Unlike our previous Reach-out to ENhancE Wellness among older cancer survivors (RENEW) RCT that found significant improvements in both physical and mental QOL with a home-based diet and exercise weight management intervention among 641 breast, prostate and colorectal cancer survivors, the current RCT only detected significant improvements in physical QOL, which it observed in both study arms. A probable explanation for this discrepancy was the smaller sample size, as the DAMES study which also has a more modest sample size, likewise was unable to detect changes in this outcome.                                                                                                                                                                                     

  1. A suggestion: The tables are too wide with too much data distributed across rows. It makes the reading difficult, so I wonder if there is an alternate formatting option or perhaps leave some data out to help the relevant data be seen more easily. As per the suggestions of the reviewer (as well as other reviewers), we have removed some extraneous information and have reformatted the tables. 

Round 2

Reviewer 2 Report

I still think that focusing on the central outcomes might have improved clarity of the paper. Nevertheless, this is up to the authors´ decision. From a scientific point of view, I have no more comments to make.